# Hypothermia protects against ventilator-induced lung injury by limiting IL-1β release and NETs formation

**Nobuyuki Nosaka[1,2†], Vanessa Borges[1,2†], Daisy Martinon[1,2], Timothy R Crother[1,2], Moshe Arditi[1,2,3], Kenichi Shimada[1,2]***

[1]Department of Pediatrics, Division of Infectious Diseases and Immunology, Guerin Children's at Cedars-Sinai Medical Center, Los Angeles, United States; [2]Infectious and Immunologic Diseases Research Center (IIDRC) and Department of Biomedical Sciences, Cedars-Sinai Medical Center, Los Angeles, United States; [3]Smidt Heart Institute, Cedars-Sinai Medical Center, Los Angeles, United States

## eLife Assessment

This study provides a comprehensive exploration of the role of hypothermia of mitigating IL-1β induction and NETosis in the context of lung injury induced by mechanical ventilation. The data are **convincing**, and the study is **important** for the field.

**Abstract** Although mechanical ventilation is a critical intervention for acute respiratory distress syndrome (ARDS), it can trigger an IL-1β-associated complication known as ventilator-induced lung injury. In mice, we found that lipopolysaccharide (LPS) and high-volume ventilation, LPS-HVV, lead to hypoxemia with neutrophil extracellular traps (NETs) formation in the alveoli. Furthermore, $Il1r1^{-/-}$ LPS-HVV mice did not develop hypoxemia and had reduced NETs, indicating that IL-1R1 signaling is important for NETs formation and hypoxemia. Therapeutic hypothermia (TH) is known to reduce the release of inflammatory mediators. In LPS-HVV mice, TH (32°C body temperature) prevented hypoxemia development, reducing albumin leakage, IL-1β, gasdermin D (GSDMD), and NETs formation. We also observed that LPS-primed macrophages, when stimulated at 32°C with ATP or nigericin, release less IL-1β associated with reduced GSDMD cleavage. Thus, hypothermia is an important modulating factor in the NLRP3 inflammasome activation, IL-1β release, and NETs formation, preventing LPS-HVV-induced acute respiratory failure.

## Introduction

Acute respiratory distress syndrome (ARDS) is a serious pulmonary disorder defined by the onset of non-cardiogenic pulmonary edema and hypoxemia (*Force et al., 2012*). While it has been more than 50 years since ARDS was first described in the literature, ARDS is still a major cause of respiratory failure in critically ill patients (*Matthay et al., 2019*). Positive-pressure mechanical ventilation (MV) has become an essential supportive strategy for the management of ARDS (*Del Sorbo et al., 2017*). However, with the onset of MV as a treatment, it was eventually understood that MV itself can cause and aggravate lung injury, a condition termed ventilator-induced lung injury (VILI) (*Rittayamai and Brochard, 2015*). Although the protective ventilation strategy using low tidal volume has been proven to decrease mortality of ARDS (*Pham and Rubenfeld, 2017*), the mortality rate of ARDS remains as high as 40% (*Bellani et al., 2016*). Non-injurious MV was reported to be able to still activate

***For correspondence:**
kenichi.shimada@cshs.org

[†]These authors contributed equally to this work

**Competing interest:** The authors declare that no competing interests exist.

**eLife digest** Patients suffering from acute respiratory distress syndrome, a serious illness that can affect people with existing conditions, usually require ventilators to assist them with breathing. However, ongoing inflammation and changes to their conditions can complicate this breathing support, sometimes causing ventilator-induced lung injury (VILI).

VILI can lead to fluid accumulation in the lungs, tissue damage and abnormally low oxygen levels in the blood. The condition has been linked to immune cells, called neutrophils, which release sticky webs as a defense against invading microorganisms. However, together with a mediator of the inflammatory immune response, the cytokine IL-1β, these neutrophil extracellular traps, or NETs, can worsen inflammation and increase damage to the lungs.

Scientists have been searching for ways to mitigate this damage, and one promising strategy is therapeutic hypothermia, a controlled method of lowering body temperature. However, it has been unclear if cooling can affect the release of IL-1β and the formation of NETs.

To find out more, Nosaka et al. used a mouse model of acute respiratory distress syndrome and VILI. The researchers injected mice with lipopolysaccharide to mimic the clinical setting of ventilated patients under infection-induced septic shock, or saline solution as a control, and exposed them to high ventilation. They then measured blood pressure, blood oxygen levels and immune factors in the lung. This revealed that IL-1β boosts NET formation, which clogged the mice's lungs and induced acute lung injury. Cooling the mice's bodies to 32°C significantly reduced lung damage. It also lowered IL-1β levels and prevented the formation of NETs, thus protecting the lungs.

Further tests on immune cells showed that hypothermia slowed key steps in inflammation, which reduced harmful immune responses. These results suggest that lowering the body temperature could be a simple and effective way to protect lungs when ventilators are needed, which could be beneficial in the treatment of conditions such as acute respiratory distress syndrome, COVID-19 or other severe lung diseases.

Therapeutic hypothermia could become an easy, non-invasive way to protect the lungs of critically ill patients and improve hospital care. However, before this treatment can be widely used, clinical trials in humans are needed to confirm its safety and effectiveness.

proinflammatory signals in the lung (*Gharib et al., 2009*). Understanding the mechanisms of VILI should lead to novel strategies to further reduce mortality in ARDS (*Rittayamai and Brochard, 2015*).

Neutrophils are key players in the development of ARDS and VILI, as their infiltration is a hallmark of lung injury progression (*Grommes and Soehnlein, 2011*). Over the past decade, neutrophil extracellular traps (NETs) have been the subject of intense research, with many advances made (*Mutua and Gershwin, 2021*). NETs are a neutrophil-derived meshwork of chromatin fibers decorated with granule peptides and enzymes and represent a critical host defense strategy against invading microorganisms (*Castanheira and Kubes, 2019*). Intriguingly, recent studies found that NETs play a role in the pathogenesis of tissue injuries, including ARDS and VILI, either with or without infection (*Porto and Stein, 2016*), and NETs have emerged as an important player in COVID-19 pathogenesis (*Silva et al., 2022*; *Veras et al., 2020*; *Castanheira and Kubes, 2023*). However, the mechanism of NETs formation and its functional implications in alveolar space during VILI are incompletely understood.

IL-1β has been linked with many inflammatory disorders (*Arend and Guthridge, 2000*). IL-1β levels in BALF were increased in patients with ventilator-associated pneumonia (*Conway Morris et al., 2010*). Furthermore, plasma levels of IL-1β have been associated with worse outcomes in ARDS patients (*Meduri et al., 1995*). However, while how IL-1β mechanistically affects ARDS is still unknown, it seems that increased amounts of IL-1β are associated with the severity of ARDS. NOD-like receptor family pyrin domain-containing protein 3 (NLRP3) inflammasome activation and interleukin (IL)-1β release by alveolar macrophages (AMs) was identified as a mechanism for severe acute lung injury (ALI) development in a two-hit model of ARDS using lipopolysaccharide (LPS) instillation and MV (*Jones et al., 2014*). IL-1β is the most biologically active cytokine in the acute phase of ARDS. It is generally considered that mature IL-1β signals via alveolar epithelial cells, resulting in increased lung permeability and pulmonary edema (*Ganter et al., 2008*). Recent studies have demonstrated that IL-1β and NETs share gasdermin D (GSDMD) as a facilitator of their release (*He et al., 2015*; *Sollberger*

*et al., 2018*). Furthermore, IL-1β promotes NETs formation in different experimental settings, which has attracted attention as an uncovered function of IL-1β (*Mitroulis et al., 2011*; *Apostolidou et al., 2016*). However, whether IL-1 participates in NETs induction in lung injury is still unknown.

Therapeutic hypothermia (TH) has been generating interest as a promising strategy for ARDS refractory to the current evidence-based therapies (*Hayek et al., 2017*). TH has long been known to be protective against severe lung injuries clinically and experimentally (*Villar and Slutsky, 1993*; *Hayek et al., 2017*). In a porcine two-hit model induced by MV and oleic acid, TH reduced the ARDS-associated lung injury and inflammation (*Angus et al., 2022*). Additionally, one case report showed the successful use of TH for severe refractory hypoxemia in a COVID-19 patient (*Cruces et al., 2021*). This has led to a phase II clinical trial called 'cooling to help injured lungs' (CHILL) in which TH is applied in association with neuromuscular blockade in ARDS patients, including those with COVID-19 (*Shanholtz et al., 2023*). TH has strong anti-inflammatory effects, and previous studies have reported inhibition of IL-1β production under hypothermia (*Diao et al., 2020*). However, little is known about how TH affects IL-1β production and NETs formation.

In this study, we found that IL-1R1 signaling enhanced NETs formation, which contributed to the development of hypoxemia and severe ALI. In addition, we found that TH inhibited IL-1β release from macrophages, which led to less NETs formation and albumin leakage in the alveolar space in our lung injury model. These results add new insights into IL-1β signaling in VILI and ARDS, and its down-modulation by TH, which could provide new therapeutic targets.

## Results

### Hypoxemia and NETs formation in alveoli during severe ALI induced by LPS plus MV

To better understand the mechanisms underlying ARDS and VILI, we subjected C57BL/6 mice to intratracheal instillation of LPS and MV with high-volume ventilation (HVV) or low-volume ventilation (LVV). At 30 and 150 min of MV, arterial blood gases were measured, and at the end of 180 min, the animals were euthanized and the bronchoalveolar lavage fluid (BALF) was collected (*Figure 1A*). The combination of LPS and HVV (LPS-HVV) caused a prominent reduction in the partial pressure of oxygen (*Figure 1B*) in the arterial blood (PaO$_2$) of mice when compared to the other controls receiving normal saline (NS) and/or LVV instillation, indicative of lung dysfunction. The requirement of both LPS and HVV for hypoxemia development was confirmed by the strong interaction (p<0.001) between these two factors found by the statistical analysis, and it occurred without significant change in carbon dioxide partial pressure (PaCO$_2$), pH, and base excess in the arterial blood (*Figure 1—figure supplement 1A–D*). Hypoxemia was accompanied by increased neutrophil migration (*Figure 1C*) compared with NS+HVV without significant change in the number of macrophages (*Figure 1D*) as determined by flow cytometry (*Figure 1—figure supplement 2*) and in the total cell number (*Figure 1—figure supplement 1E*) in BALF. To evaluate the induction of local inflammatory responses in the alveoli, we assessed the levels of several inflammatory markers in BALF. Only LPS, but not HVV, was required for the increase of IL-6, TNFα, and CXCL1. On the other hand, only HVV was sufficient to increase the IL-18 release (*Figure 1—figure supplement 1F–I*). Although these increased inflammatory mediators do not indicate interactions between LPS and HVV in the proposed model, LPS+HVV increased albumin levels in the BALF (*Figure 1E*), indicating increased vascular permeability. This injury was associated with increased IL-1β (*Figure 1F*), IL-1α (*Figure 1G*), CXCL2 (*Figure 1H*), plasminogen (*Figure 1—figure supplement 1J*), and fibrinogen (*Figure 1—figure supplement 1K*) in the BALF. We next sought to identify if the neutrophils were activated differently between LPS+LVV and LPS-HVV since there was no significant difference in neutrophil migration between the two models (*Figure 1C*), but only LPS-HVV developed hypoxemia (*Figure 1B*). We measured the concentration of myeloperoxidase (MPO) (*Figure 1I*) and neutrophil elastase (NE) (*Figure 1J*), two neutrophil granule enzymes which are also known to contribute to the formation of NETs (*Papayannopoulos et al., 2010*; *Metzler et al., 2014*), as well as the presence of histone-DNA (*Figure 1K*) and MPO-DNA (*Figure 1L*) complexes, validated methods for estimating general cell death and NETs (*Ishii et al., 2000*; *Yoo et al., 2014*), respectively. We found that LPS-HVV significantly increased all these NETs-associated parameters, indicating the involvement of NETs in the alveolar space with the local severe ALI, resulting in hypoxemia observed in this model.

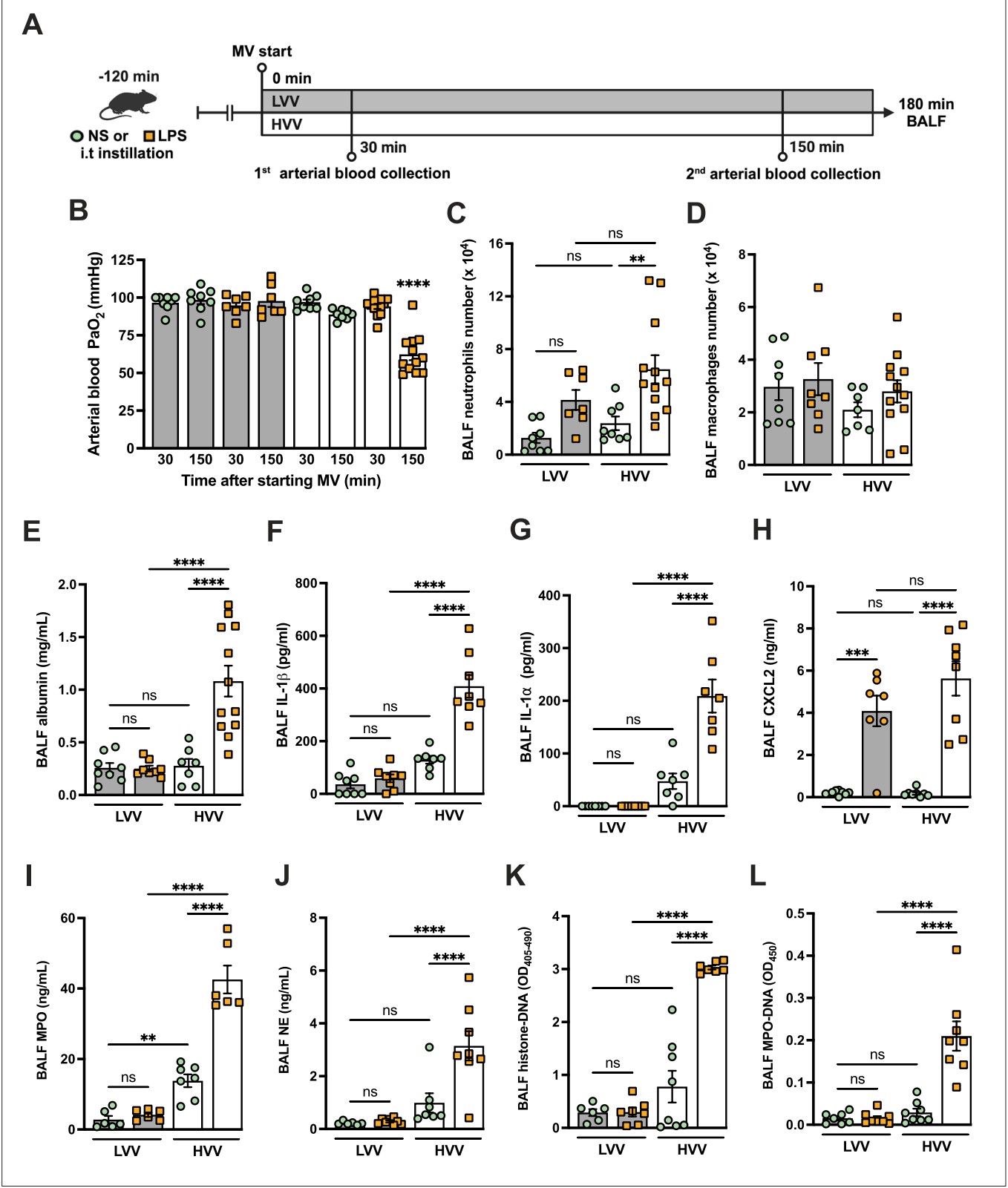

**Figure 1.** Severe acute lung injury induced by lipopolysaccharide (LPS) plus high-volume mechanical ventilation (MV) is associated with neutrophil extracellular traps (NETs) formation in the alveoli. LPS or normal saline (NS) was intratracheally instilled into C57BL/6 mice, and after 120 min, the animals were anesthetized and placed on MV for 180 min with the tidal volumes of 30 mL/kg, high-volume ventilation (HVV), or 10 mL/kg low-volume ventilation (LVV) (**A**). This panel was created using BioRender.com. Arterial blood partial pressure of oxygen (PaO$_2$) was measured at 30 and 150 min after starting

*Figure 1 continued on next page*

Figure 1 continued

MV (**B**). Absolute counts of neutrophils (**C**) and macrophages (**D**) were determined in bronchoalveolar lavage fluid (BALF). The levels of albumin (**E**), IL-1β (**F**), IL-1α (**G**), CXCL2 (**H**), MPO (**I**), and NE (**J**) in BALF collected from euthanized animals after 180 min of MV were determined by ELISA. Cell death in BALF was evaluated by measuring histone-DNA complexes (**K**). NETs formation was evaluated by the detection of MPO-DNA complex (**L**). ****, ***, and ** indicate $p<0.0001$, $p<0.001$, and $p<0.01$, respectively, determined by three-way ANOVA (**B**) and two-way ANOVA (**C–L**) followed by Tukey's multiple comparisons test; ns, nonsignificant; the absence of asterisks means nonsignificant between all the groups; **** on B indicate that the group is different from all the other groups; values are the mean ± SEM; n=7–12.

The online version of this article includes the following source data and figure supplement(s) for figure 1:

**Source data 1.** Raw numerical values for *Figure 1* plots.

**Figure supplement 1.** Severe acute lung injury induced by lipopolysaccharide (LPS) plus high-volume mechanical ventilation (MV) occurs without alteration in respiratory acidosis or alkalosis and increased plasminogen and fibrinogen levels in the alveoli.

**Figure supplement 2.** Gate strategy for alveolar neutrophils and macrophages in the two-hit model.

## Neutrophils are required for the development of severe ALI in the LPS+HVV model

To further investigate the role of neutrophils in this LPS-HVV-induced hypoxemia, we depleted neutrophils in vivo by treating C57BL/6 mice with antibodies against Ly6G (*Carr et al., 2011*) 18 hr before starting MV (*Figure 2A*). Neutrophil depletion prevented hypoxemia during LPS-HVV, while mice receiving isotype control antibody developed hypoxemia as before (*Figure 2B*). The absence of neutrophils in the alveoli was confirmed by the reduced total cell number and nearly a complete absence of neutrophils in BALF (*Figure 2C and D*). Moreover, in mice depleted of neutrophils, the number of macrophages was quite similar (*Figure 2E*). The absence of neutrophils significantly reduced the concentration of albumin and IL-6 in BALF (*Figure 2F and H*), but did not affect the amount of IL-1β and TNFα (*Figure 2G and I*). We also found higher concentrations of the chemokine CXCL2 (*Figure 2J*) in neutrophil-depleted mice. As expected, neutrophil depletion also resulted in lower MPO and NE concentrations (*Figure 2K and L*), as well as reduced detection of cell death and NETs (*Figure 2M and N*) in the BALF.

## NETs contribute to the development of severe ALI in the LPS-HVV model

To investigate the functional role of NETs in the developing hypoxemia in the LPS-HVV-induced severe ALI, we used two different approaches. In the first, we used neutrophil-specific PAD4-deficient (*Padi4*$^{Δ/Δ}$ *S100a8*$^{cre}$) mice and controls (*Padi4*$^{fl/fl}$) (*Figure 3A*). PADs are required for NETs formation (*Rohrbach et al., 2012*). Neutrophil-specific PAD4 deletion significantly inhibited the hypoxemia development compared with control mice (*Figure 3B*) but without affecting the numbers of neutrophils (*Figure 3C*) and macrophages (*Figure 3—figure supplement 1A and B*) in the alveoli. It also resulted in reduced levels of BALF albumin but did not change the IL-1β amount in the BALF (*Figure 3D and E*). PAD4 deficiency in neutrophils did not alter the levels of MPO and NE in BALF (*Figure 3—figure supplement 1C and D*) but did reduce cell death as measured by histone DNA (*Figure 3—figure supplement 1E*) and NETs formation as measured by MPO-DNA complexes (*Figure 3F*). We next treated C57BL/6 mice with DNase I (*Figure 3G*) aiming to eliminate NET structures (*Czaikoski et al., 2016*). This intervention also significantly prevented hypoxemia (*Figure 3H*) with no impact on neutrophil migration (*Figure 3I*) and number of macrophage (*Figure 3—figure supplement 1G*). It also reduced the albumin leakage (*Figure 3J*) without altering IL-1β levels (*Figure 3K*) in the alveoli. We observed that DNase I treatment resulted in similar levels of MPO and NE (*Figure 3—figure supplement 1H and I*) and resulted in efficiently reduced cell death as measured by histone DNA (*Figure 3—figure supplement 1J*) and NETs formation as measured by MPO-DNA complexes (*Figure 3L*).

## IL-1R1 signaling is required for NETs formation and severe ALI development in the LPS-HVV model

We previously reported that the activation of NLRP3 inflammasome and IL-1β are required for LPS and MV-induced two-hit model of ALI in mice (*Jones et al., 2014*; *Nosaka et al., 2020*). To confirm the participation of IL-1 receptor type 1 (IL-1R1) downstream signaling in the severe ALI development in this two-hit model, we submitted wild-type (WT) and *Il1r1*$^{-/-}$ mice to LPS-HVV (*Figure 4A*). *Il1r1*$^{-/-}$ mice

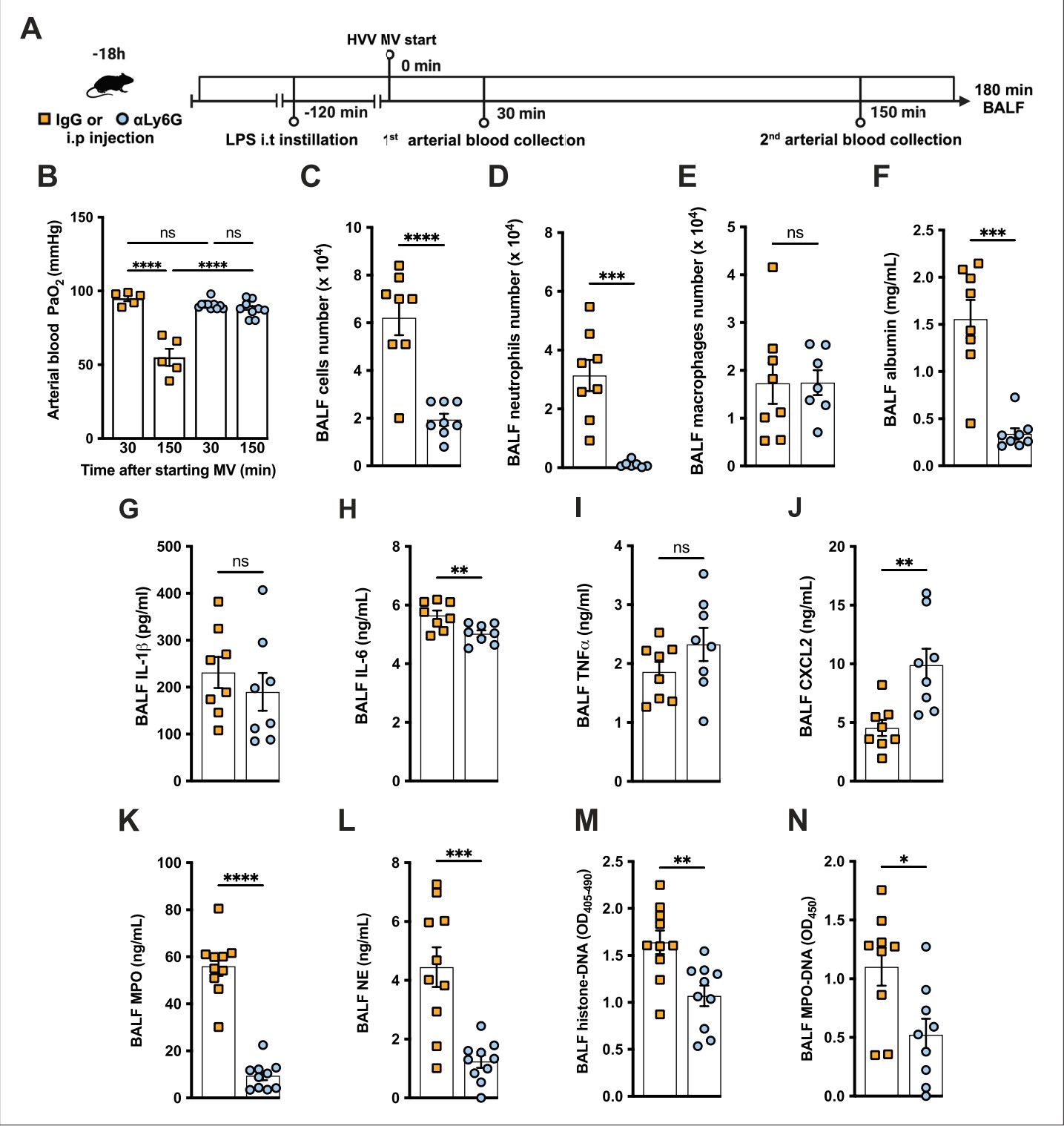

**Figure 2.** Neutrophils are required for the development of severe acute lung injury in the lipopolysaccharide (LPS)+high-volume ventilation (HVV) model. Eighteen hours before starting mechanical ventilation (MV), the anti-neutrophil monoclonal antibody (αLy6G [1A8]) or the control IgG was administered i.p. to C57BL/6 mice. Sixteen hours later, LPS was instilled i.t. in the mice, and after 120 min, they were anesthetized and placed on HVV for 180 min (**A**).This panel was created using BioRender.com. Arterial blood partial pressure of oxygen (PaO₂) was measured at 30 and 150 min after starting MV (**B**). Absolute counts of total cells (**C**), neutrophils (**D**), and macrophages (**E**) in bronchoalveolar lavage fluid (BALF). The concentration of albumin (**F**), IL-1β (**G**), IL-6 (**H**), TNFα (**I**), CXCL2 (**J**), MPO (**K**), and NE (**L**) was measured in the BALF by ELISA. Cell death and neutrophil extracellular traps (NETs) formation in the BALF were evaluated by histone-DNA (**M**) and MPO-DNA (**N**) respectively. ****, ***, **, and * indicate p<0.0001, p<0.001, p<0.01,

*Figure 2 continued on next page*

*Figure 2 continued*

and p<0.05, respectively, determined by two-way ANOVA followed by Tukey's multiple comparisons test (**B**), unpaired two-tailed Student's t-test (**C, E, G–N**), or Mann-Whitney test (**D, F**); ns, nonsignificant; values are the mean ± SEM; n=5–12.

The online version of this article includes the following source data for figure 2:

**Source data 1.** Raw numerical values for *Figure 2* plots.

did not develop hypoxemia, as shown with preserved $PaO_2$ values while WT mice have reduced $PaO_2$ (*Figure 4B*). Although IL1R1 deficiency did not affect the number of neutrophils and macrophages in BALF (*Figure 4C and D*), it prevented the development of edema resulting from increased vascular permeability, as measured by albumin in BALF (*Figure 4E*). As expected, IL-1β levels (*Figure 4F*) were not affected by loss of IL-1R1, demonstrating that there is no feedback loop required for IL-1β production in this model. Loss of this receptor also reduced the release of IL-6 in the BALF (*Figure 4G*) but did not alter the levels of TNFα and CXCL2 (*Figure 4H and I*). Even though the lack of IL-1R1 did not affect neutrophil migration, its loss led to a reduction in BALF MPO and NE levels (*Figure 4J and K*), as well as cell death (histone DNA) and the NETs formation as measured by MPO-DNA complexes in the BALF (*Figure 4L and M*). IL-1β has been found to promote NETs formation in some studies (*Mitroulis et al., 2011*; *Meher et al., 2018*). To confirm the participation of IL-1β in NETs formation in vitro, we used neutrophils from a variety of tissues, including bone marrow neutrophils (BMN), alveolar neutrophils (AN), circulating neutrophils (CN), and peritoneal neutrophils (PN). The purity of these neutrophils was evaluated by flow cytometry (CD45.2$^+$CD11b$^+$ Ly6G$^+$). Isolated BMN, AN, CN, and PN were approximately 88%, 99%, 86%, and 89% pure, respectively (*Figure 4—figure supplement 1A*). Moreover, AN nuclei look more segmented than the others (*Figure 4*) and present higher expression of CD11b and Ly6G (*Figure 4—figure supplement 1C–E*). Neutrophil quality was tested by the baseline of NETs formation (*Figure 4—figure supplement 2A*) with no stimulus after 4 hr of incubation. BMN and AN baseline NETs formation was up to 10%, while CN and PN presented about 20% and 60%, respectively, indicating lower stability under cell culture (*Figure 4—figure supplement 2B and C*). We then evaluated the NETs formation in response to IL-1β, LPS, and ionomycin (ION) by neutrophils under cell culture with concentration-response curves, as well as co-stimulation experiments (*Figure 5A*). With BMN and AN, we evaluated the concentration-response with single stimuli and co-stimulation of LPS or IL-1β with ION (*Figure 5B–G*). The concentration-response curves were also evaluated with CN and PN (*Figure 4—figure supplement 2D and E*). In all samples, the amounts of LPS and IL-1β used were not sufficient to induce NETs formation. Given the better stability of BMN and AN under cell culture, we used these cells for additional experiments. BMN were more resistant to ION, requiring a concentration of 10 μM to induce a 50% NETs formation response, while AN only required 3 μM (*Figure 5C and F*). The sensitivity difference between BMN and AN NETs formation may be related to their maturity state, as AN have highly segmented nuclei and higher CD11b and Ly6G expressions compared with BMN (*Figure 4—figure supplement 1B–E*). As LPS and IL-1β did not induce NETs formation by themselves, we next investigated if they could alter the response to ION. For this, neutrophils were stimulated with 10 μg/mL of LPS or 100 ng/mL of IL-1β, and 10 μM (BMN) or 3 μM (AN) of ION, and NETs formation was evaluated. We found that IL-1β, but not LPS, enhanced ION-induced NETs formation of both BMN (*Figure 5C and D*) and AN (*Figure 5F and G*). These data demonstrate that IL-1 signaling is pivotal for hypoxemia development and can modulate NETs formation in LPS+HVV ALI model.

## Hypothermia protects against LPS-HVV-induced ALI

During the course of our mouse studies, we observed that maintaining normal body temperature was important in obtaining consistent results. Several studies have proven that TH can modulate the inflammatory response controlling the release of a variety of inflammatory mediators including IL-1β (*Diao et al., 2020*). Thus, we decided to investigate whether hypothermia treatment might provide a feasible way to modulate the development of LPS-HVV-mediated severe ALI. We then subjected C57BL6 to the LPS+HVV model under controlled body temperature of 37±1°C or 32 ± 1°C, designated as normothermia and hypothermia, respectively (*Figure 6A and B*). Hypothermia provided strong protection against hypoxemia throughout the time course (*Figure 6C*). While similar neutrophil and macrophage counts were observed between the two groups (*Figure 6D and E*), hypothermia

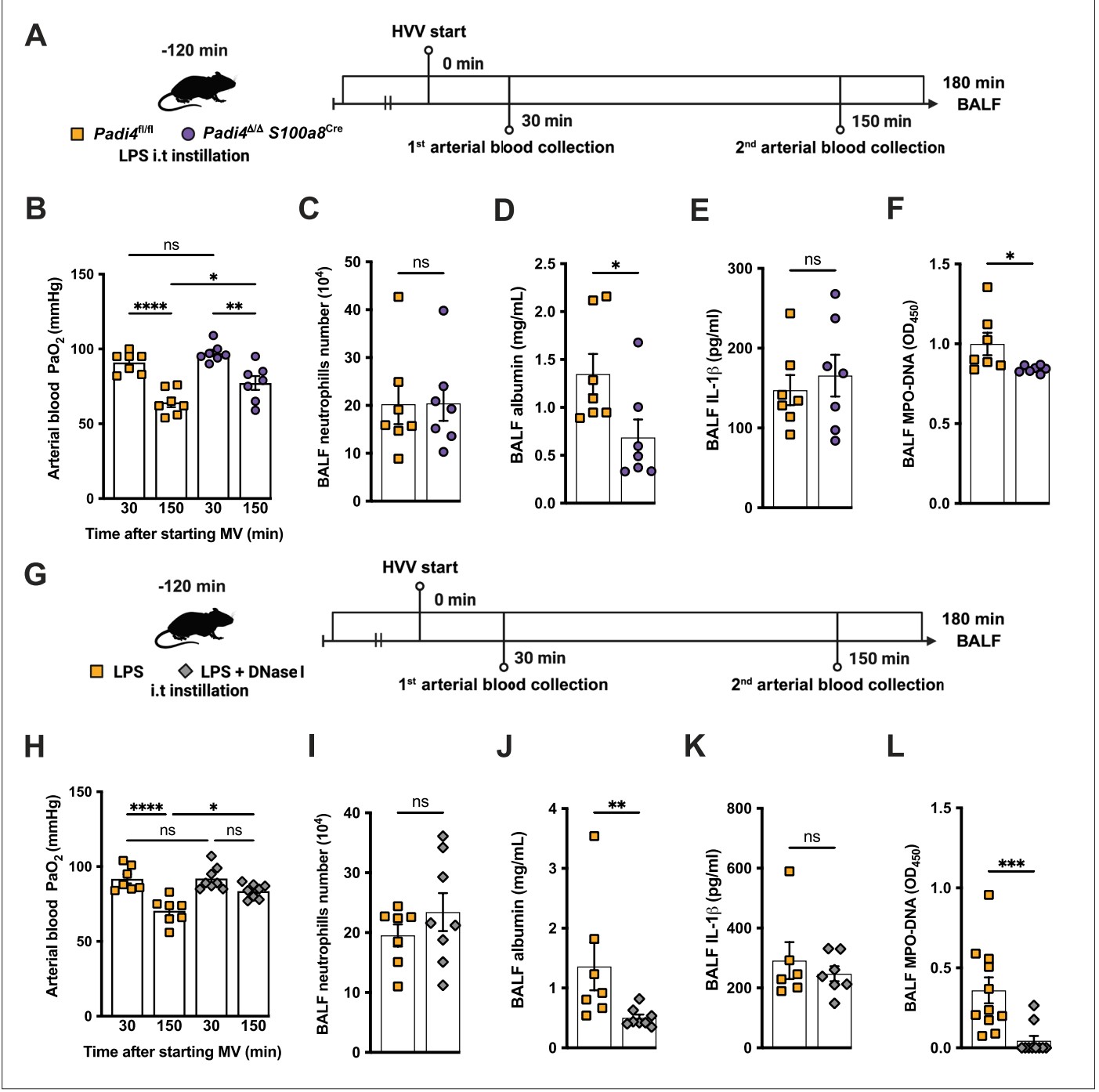

**Figure 3.** Neutrophil extracellular traps (NETs) contribute to the development of severe hypoxemia in the lipopolysaccharide (LPS)-high-volume ventilation (HVV)-induced acute lung injury (ALI). LPS was instilled to neutrophil-specific PAD4-deficient (*Padi4*$^{\Delta/\Delta}$*S100a8*$^{Cre}$) or the controls (*Padi4*$^{fl/fl}$) mice, and after 120 min, the animals were placed on mechanical ventilation (MV) for 180 min (**A**).This panel was created using BioRender.com. Arterial blood partial pressure of oxygen (PaO$_2$) was measured at 30 and 150 min after starting MV (**B**). Absolute counts of neutrophils (**C**) and the levels of albumin (**D**) and IL-1β (**E**) were measured in bronchoalveolar lavage fluid (BALF). NETs formations in BALF were evaluated by measuring MPO-DNA (**F**). LPS and DNase I were instilled i.t. to C57BL/6 mice, and mice were placed on MV (**G**).This panel waas created using BioRender.com. PaO$_2$ was measured at 30 and 150 min after starting MV (**H**). Neutrophils (**I**), albumin (**J**), IL-1β (**K**), and MPO-DNA (**L**) were measured in BALF. ****, ***, **, and * indicate $p<0.0001$, $p<0.001$, $p<0.01$, and $p<0.05$, respectively, determined by two-way ANOVA followed by Tukey's multiple comparisons test (**B, H**), unpaired two-tailed Student's t-test (**C, E, F, I**) or Mann-Whitney test (**D, J–L**); ns, nonsignificant; values are the mean ± SEM; n=7–11.

The online version of this article includes the following source data and figure supplement(s) for figure 3:

**Source data 1.** Raw numerical values for *Figure 3* plots.

*Figure 3 continued on next page*

Figure 3 continued

**Figure supplement 1.** In the lipopolysaccharide (LPS)-high-volume ventilation (HVV) model, neutrophil-specific *Padi4* deletion and DNase I treatment reduce the levels of histone-DNA complexes in the alveoli without altering the macrophage number and the levels of myeloperoxidase and neutrophil elastase.

resulted in reduced levels of BALF albumin, IL-1β, IL-6, TNFα, MPO, and NE (*Figure 6F–K*). GSDMD-mediated pore formation impacts not only IL-1β release by macrophages but was also described to play a role in NETs formation (*Sollberger et al., 2018*). We therefore evaluated the soluble GSDMD concentration in the BALF by ELISA (*Figure 6L*) and found it significantly diminished in mice ventilated under hypothermia. We observed that hypothermia also significantly inhibited the presence of cell death and NETs formation in the BALF in mice subjected to LPS-HVV (*Figure 6M and N*). Finally, neither alkalosis (blood pH>7.45) nor acidosis (blood pH<7.35) was observed during hypothermia treatment (*Figure 6—figure supplement 1A and B*).

## Hypothermia inhibits macrophage IL-1β release by modulating NLRP3 inflammasome-induced GSDMD cleavage

Since we found that hypothermia inhibited two-hit-induced acute respiratory failure with reduced IL-1β in the airways, we next evaluated the ability of bone marrow-derived macrophages (BMDMs) to release IL-1β under hypothermia. BMDMs were primed with LPS for 3 hr at 37°C, incubated at 37°C or 32°C for 30 min prior to adenosine triphosphate (ATP) or nigericin (NIG) treatment and incubated for another 30 min (*Figure 7A*). Macrophages incubated at 32°C released significantly less IL-1β compared with those incubated at 37° (*Figure 7B and E*). The mechanism by which hypothermia inhibits IL-1β release seems to be independent of caspase-1 activation, as there was no difference in caspase-1 activity assay by FLICA between 37°C or 32°C treated macrophages (*Figure 7C and D*), but we found that hypothermia resulted in reduced caspase-1 release in the supernatant (*Figure 7E*). Cleavage of GSDMD is a late limiting step for inflammasome-mediated IL-1β release because its N-terminal fragment forms pores on macrophages' plasma membrane where the intracellular cytokine crosses into the extracellular compartment (*He et al., 2015*). Thus, we investigated the effect of 37°C or 32°C temperature on GSDMD expression and cleavage in BMDMs by immunofluorescence and observed that at 32°C BMDMs express less GSDMD with reduced GSDMD cleavage (*Figure 7F–H*). Furthermore, 32°C treatment inhibited GSDMD secretion. These data corroborated our observation that while hypothermia inhibited the IL-1β release in the supernatant, there was a concomitant accumulation of unreleased mature IL-1β in the macrophages incubated at 32°C (*Figure 7E*). Another mechanism by which the NLRP3 inflammasome activity is regulated is by autophagy (*Nosaka et al., 2020*). To identify whether hypothermia regulates IL-1β release by inducing autophagy, we isolated BMDMs from *Atg16l1*^fl/fl or *Atg16l1*^fl/fl *Lyz2*^Cre+ mice and repeated the previous experiment. Hypothermic condition (32°C temperature) again significantly inhibited IL-1β release even in macrophages with impaired autophagy, suggesting that the hypothermia effect is independent of autophagy (*Figure 7—figure supplement 1*) and predominantly through diminished activation of GSDMD.

Finally, we investigated the impact of hypothermia on NETs formation in vitro. BMN or AN were preconditioned at 37°C or 32°C 1 hr prior to stimulation, and then stimulated for 4 hr with 10 or 3 μM of ION, respectively (*Figure 7—figure supplement 2A*). We observed significantly reduced NETs formation in both BMN and AN stimulated at 32°C when compared to experiments conducted at 37°C (*Figure 7—figure supplement 2B and C*). Taken together, these data demonstrated that hypothermia could be a therapeutic strategy to modulate both IL-1β release and NETs formation for preventing the development of severe acute respiratory failure.

## Discussion

Recent studies suggest that excessive NETs formation plays an important role in the development of multiple diseases, including ALI (*Porto and Stein, 2016*). We have previously reported that NLRP3 inflammasome activation and IL-1β release from AMs are required for the development of hypoxemia in a mouse model of VILI induced by LPS plus MV, and inhibition of IL-1β signaling via anakinra (IL-1RA), an IL-1 receptor antagonist, attenuates the hypoxemia in this model (*Jones et al., 2014*). However, IL-1α also signals through the IL-1R1. Thus, a second study confirmed that neither anti-IL-1α-treated

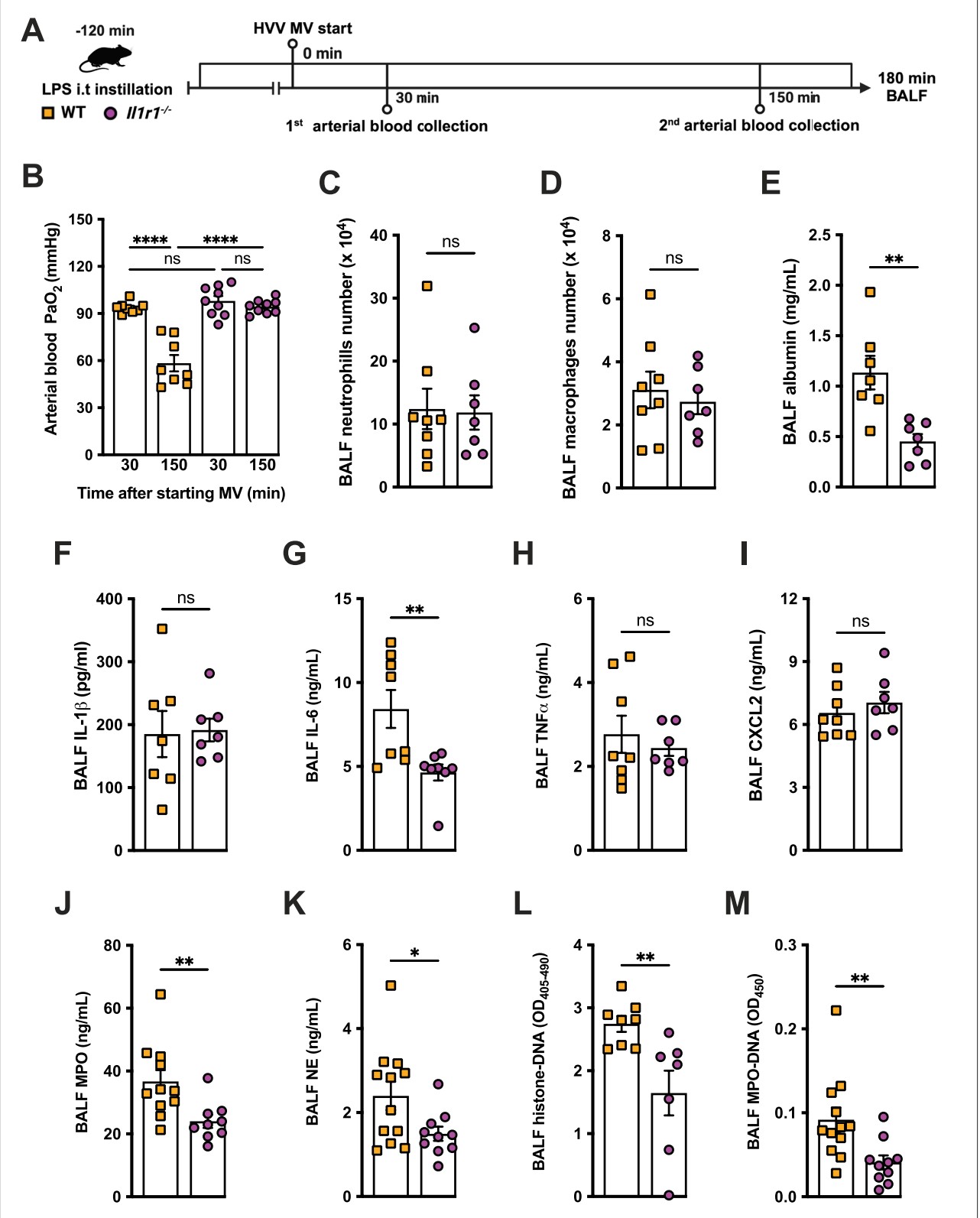

**Figure 4.** IL-1R1 signaling is required for neutrophil extracellular traps (NETs) formation in lipopolysaccharide (LPS)+high-volume ventilation (HVV)-induced acute lung injury (ALI). LPS was instilled i.t. into wild-type (WT) and *Il1r1*−/− mice, and after 120 min, the animals were anesthetized and placed on HVV for 180 min, followed by sacrifice (**A**). This panel was created using BioRender.com. Arterial blood partial pressure of oxygen was measured at 30 and 150 min after starting mechanical ventilation (MV) (**B**). Absolute counts of neutrophils (**C**) and macrophages (**D**) in bronchoalveolar lavage fluid

*Figure 4 continued on next page*

*Figure 4 continued*

(BALF) were determined. The levels of albumin (**E**), IL-1β (**F**), IL-6 (**G**), TNFα (**H**), CXCL2 (**I**), MPO (**J**), and NE (**K**) were measured in the BALF by ELISA. Cell death and NETs formation in the BALF were evaluated by measuring histone-DNA (**L**) and MPO-DNA (**M**), respectively. ****, **, and * indicate p<0.0001, p<0.01, and p<0.05, respectively, determined by two-way ANOVA followed by Tukey's multiple comparisons test (**B**), unpaired two-tailed Student's t-test (**D–F, H–M**), or Mann-Whitney test (**C, G**); ns, nonsignificant; values are the mean ± SEM; n=7–12.

The online version of this article includes the following source data and figure supplement(s) for figure 4:

**Source data 1.** Raw numerical values for *Figure 4* plots.

**Figure supplement 1.** Alveolar neutrophils (AN) present more segmented nuclei and mature CD11b and Ly6G expression.

**Figure supplement 2.** Non-stimulated circulating neutrophils (CN) and peritoneal neutrophils (PN) are more susceptible to forming neutrophil extracellular traps (NETs) compared with bone marrow neutrophils (BMN) and alveolar neutrophils (AN).

mice nor IL-1α KO mice were protected (*Nosaka et al., 2020*). Furthermore, IL-18 is not sufficient to induce hypoxemia, as saline+HVV-treated mice do not develop hypoxemia but still induce IL-18 (*Nosaka et al., 2020*). Thus, IL-1β – but not IL-1α or IL-18 – appears to play a critical role in inducing hypoxemia during LPS+HVV. In the present study, we now show that IL-1β signaling is important for NETs formation in LPS-stimulated mice undergoing MV, with supporting data demonstrating that IL-1β enhances ION-induced NETs formation in vitro. Other studies have also reported that IL-1β signaling participates in NETs formation by human neutrophils, demonstrating that single stimulus with human IL-1β is sufficient to induce NETs (*Mitroulis et al., 2011*; *Meher et al., 2018*). IL-1β plays a role in the inflammation observed in the lungs of ARDS patients, and the IL-1β level correlates with the severity of disease in these patients (*Meduri et al., 1995*; *Pugin et al., 1996*). Understanding how IL-1β modulates NETs formation in VILI further enhances the importance of IL-1β as a master cytokine in ALI pathophysiology. In an LPS-induced ALI model, NETs and their components can directly injure endothelial and alveolar epithelial cells (*Saffarzadeh et al., 2012*). In fact, histones, a component of NETs, have been described as damage-associated molecular patterns and can activate Toll-like receptors on a variety of cells, participating in organ injury development (*Li et al., 2022*). Previous to the discovery of NETs, neutrophil granule components NE and MPO were already found to cause damage to endothelial glycocalyx (*Klebanoff et al., 1993*), which also contribute to lung injury. Neutrophil-derived IL-1β was shown to be released on NETs DNA fibers (*Apostolidou et al., 2016*), which could interact directly with the alveolar capillary barrier, leading to an increase in lung epithelial and endothelial permeability (*Roux et al., 2005*). In addition to the direct role for lung injury, NETs could indirectly aggravate lung injury by inducing further IL-1β production in macrophages (*Hu et al., 2017*). Furthermore, released MPO on NETs fiber was shown to be active (*Parker et al., 2012*), which could also activate AMs (*Grattendick et al., 2002*) and lead to additional IL-1β production. Our data support and expand the premise that IL-1β plays a significant role as a driver of the vicious cycle involved in ventilation-induced ALI and ARDS.

The two-hit ALI models induced by LPS plus MV have been widely accepted and studied extensively (*Domscheit et al., 2020*). Although MV with 20 mL/kg of tidal volume has been generally accepted as clinically relevant HVV, which we have also used so far, Wilson and colleagues showed that this volume was unlikely to induce substantial lung overstretch in mice (*Wilson et al., 2012*). In this study, we slightly modified our former model, using 30 mL/kg of tidal volume, to have higher volume of MV. Moreover, we added 3 cm $H_2O$ positive end-expiratory pressure (PEEP) to avoid atelectrauma and to focus on injury by lung overstretch, known as volutrauma (*Wakabayashi et al., 2014*). We confirmed that both LPS and HVV were required for the development of ALI in our model, using control mice treated with NS and/or LVV. The two-hit requirement in our model resembles the typical clinical ARDS-developing scenario that involves pneumonia and MV, as well as the two-signal for IL-1β production and release. LPS works as the primary signal to induce pro-IL-1β production, and HVV activates the NLRP3 inflammasome and induces IL-1β release in mouse AMs (*Wu et al., 2013*). Indeed, we found that IL-1β was prominently higher in BALF of LPS+HVV mice and was the main driver of our lung injury model.

Only a few studies have been focused on NETs formation as an important player in the pathogenesis of VILI (*Rossaint et al., 2014*; *Yildiz et al., 2015*; *Li et al., 2017*). However, NETs are well described as a mechanism in severe COVID-19 infection (*Li et al., 2023*) and ARDS pathogenesis (*Scozzi et al., 2022*). We found that both inhibiting NETs formation by abolishing NETs by DNase I

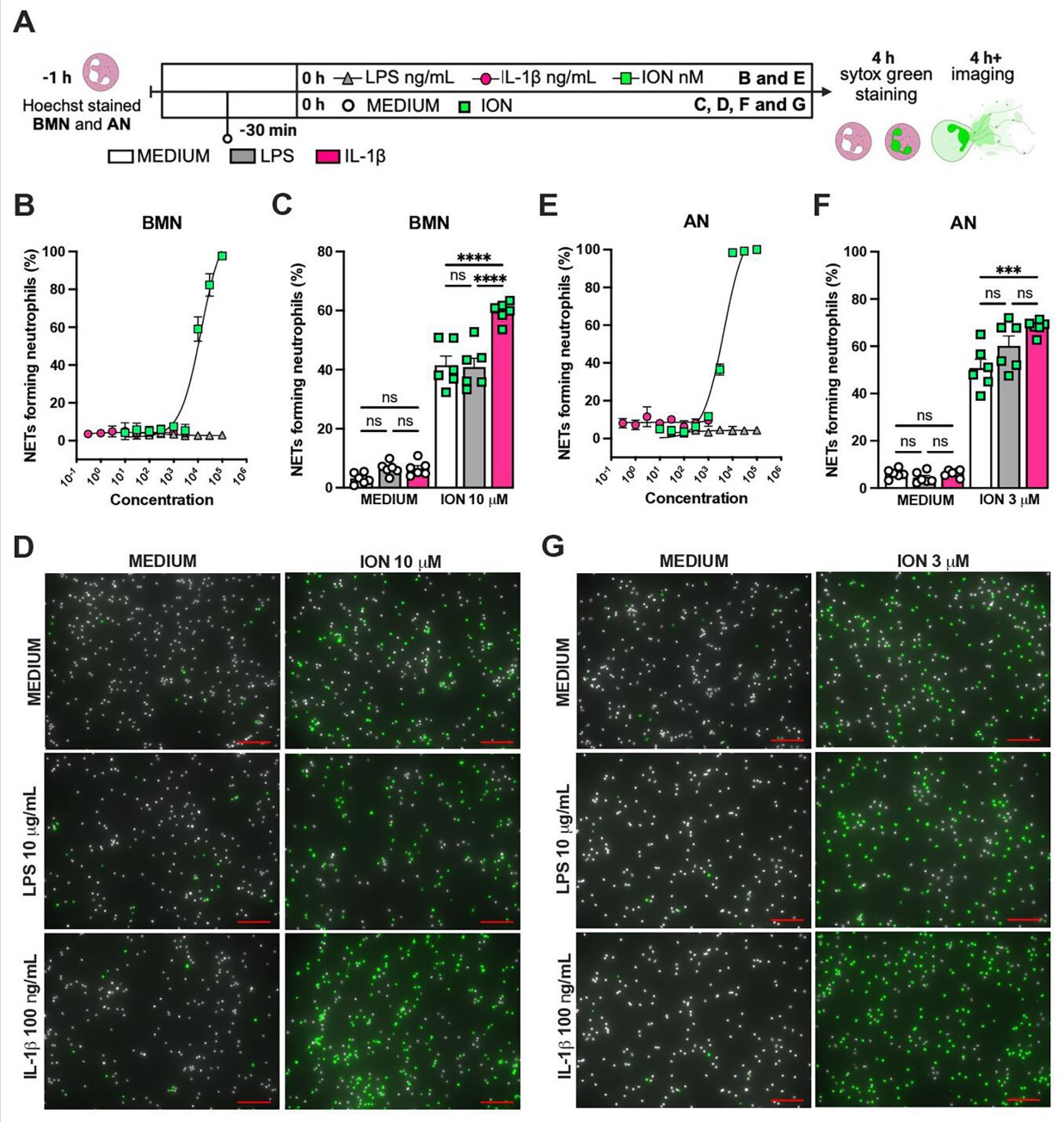

**Figure 5.** IL-1β enhances ionomycin (ION)-induced neutrophil extracellular traps (NETs) formation in vitro. Hoechst-stained bone marrow neutrophils (BMN) or alveolar neutrophils (AN) were incubated for 1 hr prior to stimulation at 37°C. At time zero, the different stimuli were added, and the cells were incubated for 4 hr, then stained with SYTOX green, and the images were captured under microscope (**A**). This panel was created using BioRender.com. BMN (**B**) and AN (**E**) were stimulated with several concentrations of lipopolysaccharide (LPS, 30–100,000 ng/mL), IL-1β (0.3–1000 ng/mL), and ION (10–100,000 nM). For combined stimulation, BMN (**C and D**) and AN (**F and G**) were first incubated with LPS or IL-1β 30 min prior to ION. The NETs forming neutrophils were analyzed as elongated shaped SYTOX Green-positive cells and expressed as percentage (%). The SYTOX Green- and Hoechst-positive cells are represented by green and white colors, respectively, on the representative images. Scale bars: 100 μm. ****, ***, and ** indicate p<0.0001, p<0.001, and p<0.01, respectively, determined by two-way ANOVA followed by Tukey's multiple comparisons test; ns, nonsignificant; values are the mean ± SEM; representative of three independent experiments.

The online version of this article includes the following source data for figure 5:

**Source data 1.** Raw numerical values for *Figure 5* plots.

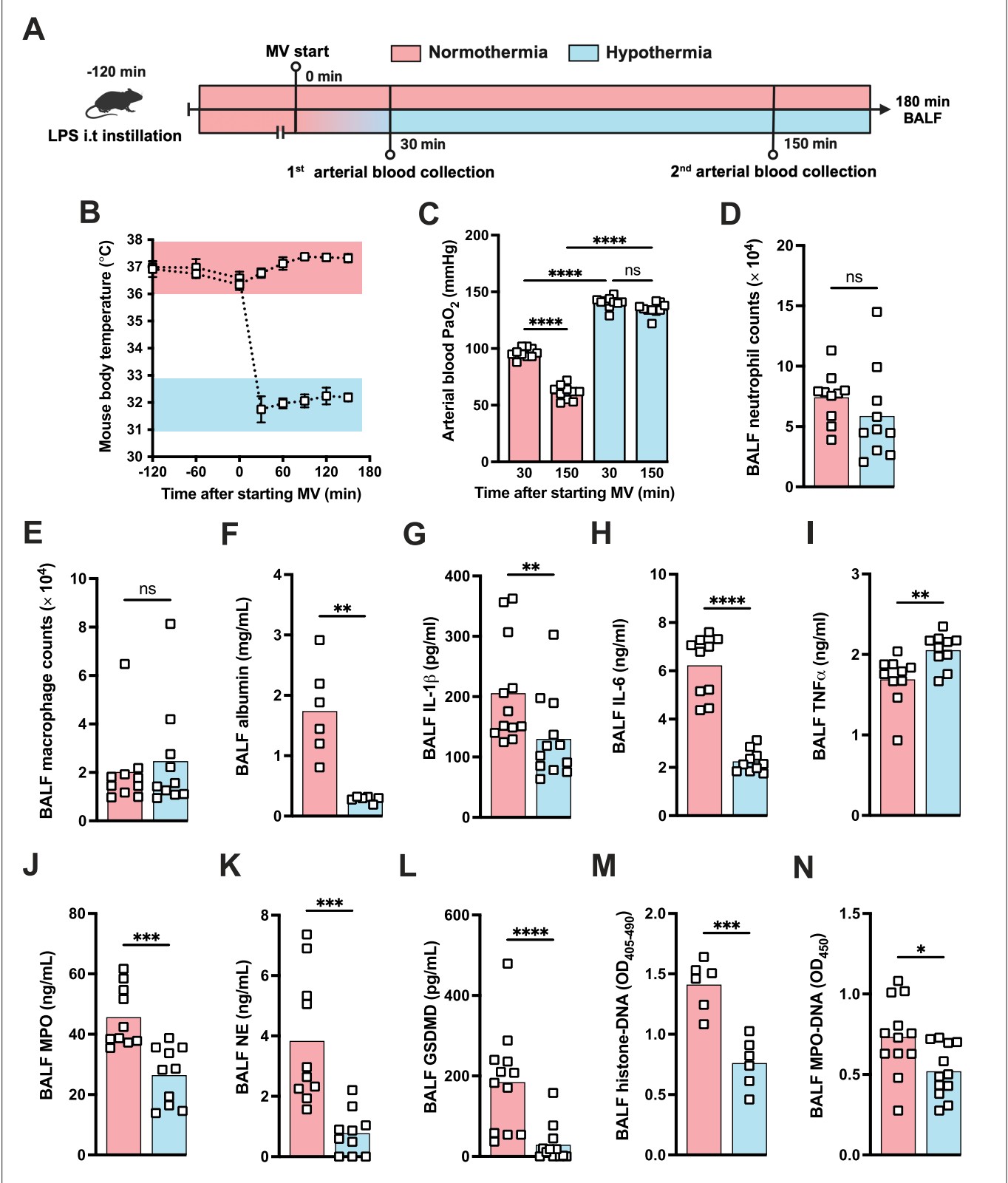

**Figure 6.** Hypothermia protects against lipopolysaccharide (LPS)+high-volume ventilation (HVV)-induced severe acute lung injury by controlling IL-1β, gasdermin D (GSDMD), and neutrophil extracellular traps (NETs) in the alveoli. LPS was instilled i.t. to C57BL/6 mice, and after 120 min, the animals were anesthetized and placed on HVV for 180 min under controlled body temperature of 37±1°C or 32±1°C, designated as normothermia and hypothermia, respectively (**A**). This panel was created using BioRender.com. The body temperature for each group was monitored (**B**). Arterial blood partial pressure

*Figure 6 continued on next page*

*Figure 6 continued*

of oxygen was measured at 30 and 150 min after starting mechanical ventilation (MV) (**C**). Absolute counts of neutrophils (**D**) and macrophages (**E**) in the bronchoalveolar lavage fluid (BALF) collected from euthanized animals after 180 min of MV. The levels of albumin (**F**), IL-1β (**G**), IL-6 (**H**), TNFα (**I**), MPO (**J**), NE (**K**), soluble GSDMD (**L**) in the BALF were determined by ELISA. Cell death and NETs formation in the BALF were evaluated by histone-DNA (**M**) and MPO-DNA (**N**), respectively. ****, ***, **, and * indicate p<0.0001, p<0.001, p<0.01, and p<0.05, respectively, determined by two-way ANOVA followed by Tukey's multiple comparisons test (**C**), unpaired two-tailed Student's t-test (**D, J, K, M, N**), or Mann-Whitney test (**E–I, L**); ns, nonsignificant; values are the mean ± SEM, n=6–12 mice/group.

The online version of this article includes the following source data and figure supplement(s) for figure 6:

**Source data 1.** Raw numerical values for *Figure 6* plots.

**Figure supplement 1.** Hypothermia protects against lipopolysaccharide (LPS) plus high-volume mechanical ventilation (MV)-induced severe acute lung injury without respiratory acidosis or alkalosis.

treatment and attenuating LPS+HVV-induced hypoxemia. Previous reports showed increased NETs in systemic circulation or in the lungs in a single-hit VILI mice model, which was markedly reduced by DNase treatment, resulting in attenuated lung injury (*Rossaint et al., 2014*; *Li et al., 2017*). In another study, Yildiz and colleagues also investigated NETs formation in the lung tissue of a two-hit VILI mice model (*Yildiz et al., 2015*). However, their model differed from ours as the authors used (20 mL/kg versus 30 mL/kg) as tidal volume, with no PEEP, for their 4 hr of MV (versus 3 hr in our study) (*Yildiz et al., 2015*). These investigators found slightly increased IL-1β levels, as well as high concentrations of DNA and citrullinated histone-h3, indirect measurement of NETs, in BALF upon LPS and MV, but DNase treatment was not sufficient to inhibit hypoxemia (*Yildiz et al., 2015*). They also did not observe an attenuation in NETs formation in response to IL-1RA treatment in their model, while we found that LPS+HVV in *IL-1R1[-/-]* mice results in reduced levels of NETs in the alveoli. These observations support the idea that IL-1β may drive more pathogenic NETs formation in LPS+HVV that leads to severe hypoxemia. Given that the role of neutrophils and types of NETs are still not clear in ALI/ARDS, further studies are clearly warranted.

A striking finding in our study was that hypothermia was protective in an LPS+HVV-mediated mouse model of VILI, preventing severe hypoxemia. Not only did hypothermia treatment in our model inhibit IL-1β release, but it also prevented NETs formation. A previous study demonstrated that hypothermia is not protective in a single-hit VILI mouse model; however, it corroborates our data by showing that hypothermia did inhibit IL-1β release in the alveoli without altering neutrophil migration (*Faller et al., 2010*). However, in a rat model of LPS-induced ALI, hypothermia inhibited neutrophil migration to the alveoli (*Lim et al., 2003*). Several studies reported lower IL-1β levels in BALF from hypothermia-treated animals subjected to LPS-induced lung injury (*Lim et al., 2003*; *Hong et al., 2005*), which is consistent with our results. Lim and colleagues also reported that hypothermia inhibits LPS-induced nuclear factor κB activation in the lungs and in AMs stimulated ex vivo (*Lim et al., 2004*). Another study reported that hypothermia attenuated the expression of caspase-1 in traumatic brain injury in rats, with reduced mature IL-1β and caspase-1 in the cerebral cortex (*Tomura et al., 2012*). In our study, we found that caspase-1 activity was unaltered in BMDMs cultured under hypothermia, but that IL-1β release was impaired and associated with lower GSDMD expression and cleavage. It has been broadly proposed that the cleaved N-terminus GSDMD can form oligomeric pores in the plasma membrane and play a key role in IL-1β secretion (*Zou et al., 2021*). While LPS-dependent palmitoylation was proposed as a requirement for N-terminus GSDMD pore formation in macrophages (*Balasubramanian et al., 2024*), the N-terminus GSDMD plasma membrane translocation and pore formation mechanisms are still poorly understood. Indeed, GSDMD binds to mitochondria outer membrane (*Yu et al., 2022*; *Miao et al., 2023*) and nuclei (*He et al., 2023*) besides plasma membrane. Furthermore, several studies have shown that soluble GSDMD is detected in culture supernatant or body fluids (*Karmakar et al., 2020*; *Nagai et al., 2021*; *Silva et al., 2022*), but our study may be the first reporting GSDMD detection in the BALF in LPS plus MV-induced severe ALI. Supporting this, GSDMD can participate in the host defense by binding to pathogen membranes, potentially forming cytotoxic pores (*Lieberman et al., 2019*). We hypothesized that hypothermia would not affect pro-IL-1β production but inhibit inflammasome activation in macrophages. We found that both immature and mature IL-1β were stuck inside the macrophages with activated caspase-1, associated with a significant reduction in mature GSDMD.

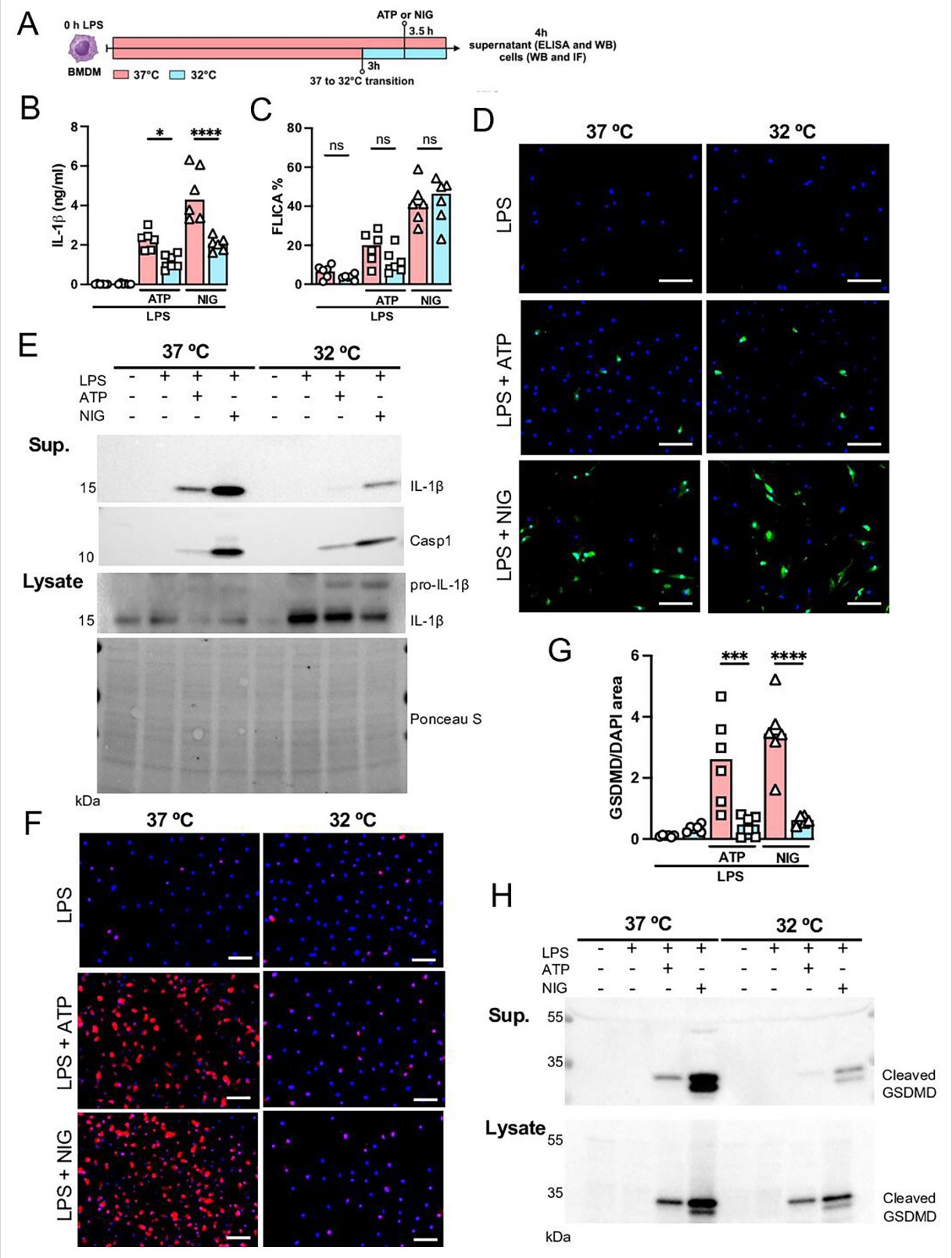

**Figure 7.** Hypothermia inhibits macrophage IL-1β release by modulating NLRP3 inflammasome-induced gasdermin D (GSDMD) cleavage. Bone marrow-derived macrophages (BMDMs) were primed with lipopolysaccharide (LPS) for 3 hr at 37°C, then incubated at 37°C or 32°C for 30 min prior to adenosine triphosphate (ATP) or nigericin (NIG) treatments for another 30 min (**A**). This panel was created using BioRender.com. In the supernatant, IL-1β concentration was determined by ELISA (**B**). The supernatant (**C**) and cell lysate (**D**) were used for western blotting (WB) analysis, and the resulting

*Figure 7 continued on next page*

*Figure 7 continued*

membranes were stained for IL-1β, caspase-1, and GSDMD (**C**). WB analysis was also made in the cell lysate, where we investigated the expression of IL-1β and GSDMD. The protein distribution in the cell lysate samples was certified by Ponceau S staining. BMDMs were stained with anti-GSDMD (red) and DAPI (blue), as shown in the representative images. The GSDMD area was analyzed and normalized by DAPI area (**E**). To evaluate caspase-1 activity, the cells were stained with FAM-YVAD-FMK FLICA and analyzed under the microscope (**F**). Scale bars: 50 µm. \*\*\*\*, \*\*\*, and \* indicate p<0.0001, p<0.001, and p<0.05, respectively, determined by two-way ANOVA followed by Tukey's multiple comparisons test; ns, nonsignificant; values are the mean ± SEM; representative of three independent experiments.

The online version of this article includes the following source data and figure supplement(s) for figure 7:

**Source data 1.** Annotated western blot images corresponding to *Figure 7E and H*.

**Source data 2.** Original, uncropped western blot images used for *Figure 7E and H*.

**Source data 3.** Raw numerical values for *Figure 7* plots.

**Figure supplement 1.** Hypothermia-induced NLRP3 inflammasome inhibition is independent of autophagy.

**Figure supplement 2.** Hypothermia inhibits neutrophil extracellular traps (NETs) formation in vitro.

One limitation of our study is that the data includes only young male mice. However, despite this limitation, we believe these results to be generally applicable as human studies have found that mortality in patients with ARDS does not differ between sexes (*Heffernan et al., 2011*). Additionally, we induced severe ALI by placing mice on HVV 2 hr after LPS administration. At this time point, pulmonary inflammation remains minimal (*Dagvadorj et al., 2015*). Moreover, LPS induces a febrile response in humans (*Ganeshan and Chawla, 2017*) and can cause either hyperthermia or hypothermia in mice, depending on the dose and ambient temperature (*Rudaya et al., 2005*). Mice and humans also differ in their sensitivity to LPS (*Ganeshan and Chawla, 2017*), highlighting limitations when translating these findings to human sepsis or infection responses. It is also important to distinguish between physiological hypothermia (just below 36°C) and TH (typically 32–34°C). While physiological hypothermia is a recognized occurrence in humans with severe infections, it is worth exploring whether TH serves as a protective response or if maintaining normothermia to hyperthermia has detrimental effects.

In summary, we developed a two-hit model of severe ALI and ARDS in which both LPS and HVV were required to induce hypoxemia in mice. We demonstrated that IL-1β signaling in neutrophils plays a role in NETs induction, which participates in the development of severe lung injury and hypoxemia. Both IL-1β and NETs have been widely implicated as having a key role in the development of ALI. This research adds specific information about the mechanisms by which severe hypoxemia observed in ARDS may be associated with IL-1β effects in lung and NETs formation. These observations suggest that the inflammasome pathway and its downstream mediators, such as IL-1β and NETosis, may be effective therapeutic targets and could be downmodulated by hypothermia during management of ARDS.

## Materials and methods
### Mice

C57BL/6, *S100a8*^Cre^, *Il1r1*^-/-^, *Lyz2*^Cre^ mice on C57BL/6 background were purchased from Jackson Laboratories (Bar Harbor, ME, USA). *Padi4*^fl/fl^ mice were provided by Dr. Kelly Mowen (Scripps Research, San Diego, CA, USA). *Atg16l1*^fl/fl^ mice were provided by Dr. Shih (Cedars-Sinai Medical Center, Los Angeles, CA, USA). All in vivo experiments were performed in mice at 8–12 weeks of age. All animal studies presented here have been approved by the Institutional Animal Care and Use Committee of the Cedars-Sinai Medical Center. All rodent experimental procedures were conducted under approved Institutional Animal Care and Use Committee protocols.

### LPS plus MV two-hit ALI model

Male mice were anesthetized with isoflurane (Piramal Healthcare, Bethlehem, PA, USA), and orotracheally intubated with an intravenous catheter (BD Insyte Autoguard, 20GA 1.00 in., Becton Dickinson Infusion Therapy Systems Inc, Sandy, UT, USA). LPS from *Escherichia coli* (LPS-EB Ultrapure, tlrl-3pelps, Invivogen) was diluted in sterile NS (0.9% sodium chloride, NS) to a concentration of 0.1 mg/mL, and 2 microliters per gram of body weight (µL/g) of LPS or NS were administered intrathracheally (i.t.)

to mice. Two hours after LPS administration, the mice were intraperitoneally (i.p.) anesthetized with ketamine (Ketaved Ketamine HCl, NDC 50989-161-06, Vedco) and dexmedetomidine (Dexdomitor dexmedetomidine hydrochloride, 122692-5, Zoetis) mixture prepared in NS, 50 and 1.0 milligrams per kilogram of body weight (mg/kg), respectively, and orotracheally re-intubated with attention to catheter insertion length and ventilated using a small animal ventilator system (VentElite, Harvard Apparatus) for 180 min with a tidal volume of 10 milliliter per kilogram of body weight (mL/kg) in a respiratory rate (RR) of 150 breaths per minute (bpm), LVV, or 30 mL/kg of body weight in an RR of 35 breaths per minute, HVV, and 3 centimeters of water ($cmH_2O$) for PEEP. For hemodynamic support, 500 µL of sterile phosphate-buffered saline (PBS) were given to each mouse at the onset of MV. A complementary dose of 25 mg/kg of ketamine and 0.5 mg/kg of dexmedetomidine was administered i.p. 90 min after starting MV, or earlier as needed. Body temperature was maintained at either 37 ± 1°C or 32 ± 1°C using a heating pad (Heated Hard Pad, Hallowell EMC) and measured rectally by using a temperature probe (08D2, DeltaTrack).

## Arterial blood gas analysis

The arterial blood was collected from anesthetized mice via tail artery by nicking the ventral side of the tail with a blade. Approximately 100 µL of whole blood was collected using a Heparinized Micro-Hematocrit Capillary Tube (Fisherbrand). Arterial blood gas was analyzed at 30 and 150 min after starting MV using i-STAT1 Analyzer and the i-STAT G3+Cartridges (Abbott, IL, USA), which provided the values of partial pressure of oxygen ($pCO_2$) and partial pressure of carbon monoxide ($pCO_2$), in millimeters of mercury (mmHg), pH and base excess as milliequivalent per liter (mEq/L).

## Bronchoalveolar lavage fluid

BALF was obtained after 180 min of MV with 0.5 mL of cold PBS with 2 mM of EDTA by inserting a standard disposable intravenous catheter (BD Insyte Autoguard, 20GA 1.00 in., Becton Dickinson Infusion Therapy Systems Inc) into the trachea. A small portion of BALF was stained with ViaStain AOPI staining solution, prepared in Cellometer cell counting chambers, and the cells were quantified in the Cellometer Auto 2000 (Nexcelom Bioscience, Lawrence, MA, USA). The supernatant was isolated for ELISA, and the remaining cells were stained with PE anti-mouse Ly6G (50-1276-U100, 1:200, Tonbo Bioscience), FITC anti-human/mouse CD11b (35-0112-U100, 1:200, Tonbo Bioscience), APC anti-mouse F4/80 (20-4801-U100, 1:200, Tonbo Bioscience), violetFluor 450 anti-mouse CD11c (75-0114-U100, 1:200, Tonbo Bioscience), and APC/Cyanine7 anti-mouse CD45.2 (109824, 1:200, BioLegend). The percentage of neutrophils (CD11b[+] Ly6G[+]) and macrophages (CD11c[+] F4/80[+]) in the gate of CD45.2[+] cells were determined by flow cytometry in the *Sony SA3800 spectral cell analyzer* (Sony Biotechnology) and analyzed using FlowJo software (FlowJo LLC 10.10.0, Becton Dickinson).

## ELISA measurements

Enzyme-linked immunosorbent assay (ELISA) kits were used for quantifying albumin (Mouse Albumin ELISA kit, 99-134, Bethyl Laboratories), IL-1β (IL-1 beta Mouse Uncoated ELISA Kit, 88-7013-88, Invitrogen), IL-1α (ELISA MAX Deluxe Set Mouse IL-1α, 433404, BioLegend), CXCL-2 (Mouse CXCL2/MIP-2 DuoSet ELISA, DY452, R&D Systems), MPO (Mouse Myeloperoxidase DuoSet ELISA, DY3667, R&D Systems), NE (Mouse Neutrophil Elastase/ELA2 DuoSet ELISA, DY4517-05, R&D Systems) cell death (Cell Death Detection ELISA, 11544675001, Roche Life Sciences), IL-6 (Mouse IL-6 ELISA Set, 555240, BD Biosciences), TNFα (TNF alpha Mouse Uncoated ELISA Kit, 88-7324-88, Invitrogen) CXCL-1 (Mouse CXCL1/KC DuoSet ELISA, DY453, R&D Systems), IL-18 (Mouse IL-18 Matched ELISA Antibody Pair Set, SEK50073, Sino Biological) plasminogen (Mouse PLG/Plasmin/Plasminogen ELISA Kit, LS-F10445, Life Span Biosciences), fibrinogen (Mouse Fibrinogen ELISA Kit, LS-F10440, Life Span Biosciences) and GSDMD (Mouse GSDMD ELISA Kit, ab233627, Abcam). In addition, we developed an ELISA based on MPO associated with DNA as previously described (*Caudrillier et al., 2012*) with some modifications. For the capture antibody, 800 ng/mL of anti-MPO capture mAb (Mouse Myeloperoxidase DuoSet ELISA, DY3667, R&D Systems) was coated onto 96-well plates overnight at room temperature (RT). After blocking and washing, 25 µL of BALF was added to the wells with 75 µL incubation buffer (1% BSA/PBS) and incubated for 2 hr at RT. After washing, incubation buffer containing a peroxidase-labeled anti-DNA mAb (Cell Death Detection ELISA, 11544675001, Roche Life Sciences, dilution 1:10). The plate was incubated for 1 hr at RT. After washing, the peroxidase substrate (TMB)

was added, and after 15 min at RT in the dark, the reaction was stopped by adding $H_2SO_4$ solution and the absorbance at 450 nm wavelength.

## Neutrophil isolation

Neutrophils were isolated from 8- to 12-week-old male WT C57BL/6 mice. The purification was made using two different density gradient protocols made with Percoll (Percoll, GE Healthcare, GE17-0891-01) prepared in Hanks' balanced salt solution (HBSS). The Percoll gradient 1 (PG1) consists of a three-layer gradient made with 75%, 57%, and 52% of Percoll, and the Percoll gradient 2 (PG2) consists of two layers, 68% and 52% of Percoll. The gradients containing 1 mL of cell suspension on the top were centrifuged for 30 min, 1500 × $g$ at RT, with soft acceleration and deceleration. Neutrophils were located above the 75% or 68% Percoll layers for PG1 and PG2, respectively. To obtain BMNs, mice were euthanized, and the rear leg bones were removed and flushed with HBSS with 2 mM of EDTA (HBSS-EDTA). The cells were centrifuged for 5 min, 450 × $g$ at RT, and the red blood cells (RBCs) were lysed by adding 10 mL of 0.2% NaCl, gently mixing for 30 s. The salt balance was recovered by adding 5 mL of 2.3% NaCl. The cells were centrifuged for 5 min, 450 × $g$ at RT, and resuspended in 1 mL of HBSS-EDTA. The cell suspension was carefully added to the PG1, and the BMN isolation was performed as above. For ANs, as described in the two-hit ALI model, mice were anesthetized with isoflurane, orotracheally intubated with intravenous catheter, and LPS 0.2 mg/kg was i.t. instilled. Three days later, BALF was obtained with 5 mL of HBSS-EDTA divided into five lavages with 1 mL of HBSS-EDTA. The cells were centrifuged for 5 min, 450 × $g$ at RT, and resuspended at 1 mL of HBSS-EDTA. The cell suspension was carefully added to PG2, and the AN were isolated. CNs were obtained from the blood collected 6 hr after LPS 0.2 mg/kg i.p. injection. The total whole blood was collected by retro-orbital bleeding, in a 5 mL tube containing 2.5 mL of EDTA (2 mg/mL in PBS). The RBCs were lysed by adding the blood (maximum 5 mL, 2.5 mL of blood+2.5 mL of EDTA) to 30 mL of 0.2% NaCl, gently mixing for 30 s. The salt balance was recovered by adding 15 mL of 2.3% NaCl. The lysis step was repeated one time in case remaining RBCs were observed in the pellet formed after 5 min of centrifugation, 450 × $g$ at RT. After total elimination of RBCs, the pellet was resuspended in 1 mL of HBSS-EDTA. The cell suspension was added to the PG2, and the CN were purified. For peritoneal neutrophil isolation, mice received i.p. injections with 1.5 mL of 3% thioglycolate, and after 16 hr, the animals were euthanized, and the peritoneal cavity was washed with 5 mL of HBSS-EDTA. The cells were centrifuged for 5 min, 450 × $g$ at RT, and resuspended in 1 mL of HBSS-EDTA. The cell suspension was added to PG2, and the neutrophils were isolated. The cells were stained with ViaStain AOPI staining solution, prepared in Cellometer cell counting chambers, quantified in the Cellometer Auto 2000, and resuspended in RPMI/neutrophils (RPMI 1640, 10-040-CV, Corning supplemented with 1% of Penicillin-Streptomycin Solution, 30-0002 CL, Corning; 2% of Fetal Bovine Serum, FB-02, Omega Scientific; and 1% of MEM Non-essential Amino Acid Solution, Sigma-Aldrich, M-7145). Neutrophil purity was evaluated by flow cytometry by staining the cells with PE anti-mouse Ly6G (50-1276-U100, 1:200, Tonbo Bioscience), FITC anti-human/mouse CD11b (35-0112-U100, 1:200, Tonbo Bioscience), and APC/Cyanine7 anti-mouse CD45.2 (109824, 1:200, BioLegend). The percentage of neutrophils (CD11b+ Ly6G+) was determined in the gate of CD45.2+ cells, analyzed using FlowJo software (FlowJo LLC 10.10.0, Becton Dickinson). The cells were also observed using cytospin (Cytospin 4, Thermo Scientific) prepared slides stained with a rapid staining of blood smear (Hemacolor Rapid staining of blood smear, 111661, Sigma-Aldrich).

## NETs quantification in vitro

$2.0×10^4$ Neutrophils were labeled with 2 µM Hoechst 33342 (Immunochemistry Technology, 639) and seeded in flat-bottom 96-well plates. We made three sets of experiments to investigate NETs formation under different conditions, using RPMI/neutrophils to prepare all the treatments. In the first set, after resting the cells for 1 hr at 37°C, the cells were treated with the following: LPS (30–100,000 ng/mL); IL-1β (recombinant mouse IL-1β protein, ab259421, Abcam) (0.3–1000 ng/mL); and ION (ionomycin calcium salt 10634, Sigma-Aldrich) (10–100,000 nM). In the second set, combined stimuli were given to BMN and AN. The cells were first rested at 37°C for 30 min, LPS (10 µg/mL) or IL-1β (100 nM) was added, and after 30 min, the cells were treated with the concentration of ION sufficient to induce about 50% of NETs formation, which is 10 µM for BMN and 3 µM for AN. In the third set, BMN and AN were placed at 37°C or 32°C for 1 hr, and then treated with ION 10 µM for BMN and 3 µM for

AN. In all of them, the cells were incubated for 4 hr at 37°C or 32°C, maintaining the initial resting temperature setting in each experiment, to allow NETs induction. The cells were then stained with 5 μM of SYTOX Green (SYTOX Green Nucleic Acid Stain, 57020, Invitrogen) prepared in sterile PBS and centrifuged for 5 min, 500 × g RT. The images were obtained on a Keyence BZ-9000 (Keyence Corporation of America) microscope at ×20 magnification. The NETs forming neutrophils percentage was evaluated manually by investigators blinded to sample identity. Quantification was based on the total cell number in the field, determined by the sum of Hoechst (white) and SYTOX Green (green) positive cells, as counted using the Keyence BioAnalyzer software (Keyence Corporation of America).

## BMDM culture

Bone marrow was obtained from 8- to 12-week-old male WT C57BL/6, *Atg16l1*[fl/fl] or *Atg16l1*[Δ/Δ] *Lyz2*[Cre] mice, and BMDMs were differentiated in RPMI/macrophages (RPMI 1640, 10-041-CV, Corning; supplemented with 1% of Penicillin-Streptomycin Solution, 30-0002-CL, Corning; 50 μM of 2-mercaptoethanol; and 10% of Fetal Bovine Serum, FB-02, Omega Scientific) containing 15% of L929 cell conditioned medium (LCM) for 7 days, supplementing the medium with extra 15% of LCM every 3 days. Cells were washed with PBS, and non-adherent cells were removed; adherent cells were then collected and seeded in a 96-well plate 1 day before stimulation. BMDMs were primed with LPS 1 μg/mL for 3 hr, followed by ATP 5 mM (adenosine 5′-triphosphate disodium salt, A2383-1G, Sigma-Aldrich) or NIG 10 μM (Nigericin sodium salt, BML-CA421-0005, Enzo Life Sciences) stimulation for 30 min at either 37°C or 32°C, and the supernatants were collected for ELISA measurements and western blotting, and the cells were lysed for western blotting. For evaluating caspase-1 activity and GSDMD expression by immunofluorescence, LPS-primed BMDMs were stimulated with 5 mM ATP for 15 min or 10 μM NIG for 30 min. For caspase-1 activity, a commercial kit (FAM-FLICA Caspase-1 Assay Kit, 98, Immunochemistry Technology) was used. Immunoblots were performed using antibodies against IL-1β (anti-IL-1 beta antibody, 2 μg/mL, ab9722; Abcam), caspase-1 (recombinant anti-pro caspase-1+p10+p12 antibody, 1:1000, ab179515; Abcam), and GSDMD (recombinant anti-GSDMD antibody, 1:1000, ab209845, Abcam). The GSDMD antibody used in the immunoblot was also used for immunofluorescence staining, which was mounted with fluorescence mounting medium with DAPI (Mounting Medium With DAPI, ab104139, Abcam). The FAM-FLICA caspase-1 images and the other immunofluorescence images were obtained on a Keyence BZ-9000 (*Keyence Corporation of America*) microscope at ×20 magnification.

## Quantification and statistical analysis

All data were analyzed using Prism 9 (GraphPad Software Inc, La Jolla, CA, USA). Normality within each group was assessed using the Shapiro-Wilk and Kolmogorov-Smirnov tests. For comparisons between two groups, the Mann-Whitney U test was used for non-normally distributed data, while the unpaired Student's t-test was applied to data that met the assumption of normality. One-way ANOVA (with a single independent factor), two-way ANOVA (with two independent factors), and three-way ANOVA (with three independent factors) were performed for comparisons involving more than two groups, followed by Tukey's post hoc test. A p-value of less than 0.05 was considered statistically significant.

## Acknowledgements

This work was supported by the National Institute of Health (NIH) grant R01-HL130353-01 to KS.

## Additional information

### Funding

| Funder | Grant reference number | Author |
| --- | --- | --- |
| National Heart, Lung, and Blood Institute | HL130353-01 | Kenichi Shimada |

| Funder | Grant reference number | Author |
| --- | --- | --- |

The funders had no role in study design, data collection and interpretation, or the decision to submit the work for publication.

## Author contributions

Nobuyuki Nosaka, Conceptualization, Formal analysis, Validation, Investigation, Visualization, Methodology, Writing – original draft, Writing – review and editing; Vanessa Borges, Data curation, Formal analysis, Validation, Investigation, Visualization, Methodology, Writing – original draft, Writing – review and editing; Daisy Martinon, Formal analysis, Validation, Investigation, Visualization; Timothy R Crother, Methodology, Writing – original draft, Writing – review and editing; Moshe Arditi, Supervision, Writing – original draft, Writing – review and editing; Kenichi Shimada, Conceptualization, Data curation, Supervision, Funding acquisition, Investigation, Methodology, Writing – original draft, Writing – review and editing

## Author ORCIDs

Nobuyuki Nosaka ⓘ https://orcid.org/0000-0002-7722-0701
Timothy R Crother ⓘ https://orcid.org/0000-0001-8465-6127
Kenichi Shimada ⓘ https://orcid.org/0000-0001-8213-5686

## Ethics

All animal studies presented here have been approved by the Institutional Animal Care and Use Committee of the Cedars-Sinai Medical Center. All rodent experimental procedure was conducted under approved Institutional Animal Care and Use Committee protocols (#6907).

Reviewer #1 (Public review): https://doi.org/10.7554/eLife.101990.3.sa1
Reviewer #2 (Public review): https://doi.org/10.7554/eLife.101990.3.sa2
Author response https://doi.org/10.7554/eLife.101990.3.sa3

# Additional files

## Supplementary files

MDAR checklist

## Data availability

All data generated or analyzed during this study are included in the manuscript and supporting files. No large-scale datasets (e.g., genomic, proteomic, or other omics data) were generated.

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
