## [Editor Report · eLife Assessment]

This study provides a comprehensive exploration of the role of hypothermia of mitigating IL-1β induction and NETosis in the context of lung injury induced by mechanical ventilation. The data are **convincing**, and the study is **important** for the field.

---

## [Referee Report · Reviewer #1 (Public review)]

Summary:

The authors found that IL-1b signaling is pivotal for hypoxemia development and can modulate NETs formation in LPS+HVV ALI model.

Strengths:

They used IL1R1 ko mice and proved that IL1R1 is involved in ALI model proving that IL1b signalling leads towards ARDS. In addition, hypothermia reduces this effect, suggesting a therapeutic option.

Comments on revised version:

The authors have addressed this Reviewer's concerns. The manuscript is much stronger in the current form and can be published.

---

## [Referee Report · Reviewer #2 (Public review)]

Summary:

The manuscript by Nosaka et al is a comprehensive study exploring the involvement of IL1beta signaling in a 2-hit model of lung injury + ventilation, with a focus on modulation by hypothermia.

Strengths:

The authors demonstrate quite convincingly that interleukin 1 beta plays a role in the development of ventilator-induced lung injury in this model, and that this role includes the regulation of neutrophil extracellular trap formation. The authors use a variety of in vivo animal-based and in vitro cell culture work, and interventions including global gene knockout, cell-targeted knockout and pharmacological inhibition, which greatly strengthen the ability to make clear biological interpretations.

Comments on revised version:

The authors have addressed my concerns/queries.

---

## [Author Response]

The following is the authors’ response to the original reviews

**Public Reviews:**

**Reviewer #1 (Public review):**
Summary:The authors found that IL-1b signaling is pivotal for hypoxemia development and can modulate NETs formation in LPS+HVV ALI model.Strengths:They used IL1R1 ko mice and proved that IL1R1 is involved in ALI model proving that IL1b signalling leads towards ARDS. In addition, hypothermia reduces this effect, suggesting a therapeutic option.

We thank the Reviewer for recognizing the strengths of our study and their positive feedback.

Weaknesses:(1) IL1R1 binds IL1a and IL1b. What would be the role of IL1a in this scenario?

Thank you for asking this question. We have addressed this in our previous paper (Nosaka et al. Front Immunol 2020;11; 207) where we used anti-IL-1a and IL-1a KO mice (Nosaka et al. Front Immunol 2020;11; 207) in our model and found that neither anti-IL-1a treated mice nor IL-1a KO mice were protected. Thus, IL-1b plays a role in inducing hypoxemia during LPS+HVV but not IL-1a. We will now add this point in our revised manuscript discussion.

(2) The authors depleted neutrophils using anti-Ly6G. What about MDSCs? Do these latter cells be involved in ARDS and VILI?

Anti-Ly6G neutrophils depletion may potentially affect G-MDSCs as well (Blood Adv 2022 Jul 29;7(1):73–86), however, we have not looked directly at G-MDSCs. If these cells were depleted we would have expected to see an increase in inflammation, which we did not. Instead, anti-Ly6G treated mice were protected. Thus, we can not comment on any presumed role of G-MDSCs in LPS+HVV induced severe ALI model that we used.

(3) The authors found that TH inhibited IL-1β release from macrophages led to less NETs formation and albumin leakage in the alveolar space in their lung injury model. A graphical abstract could be included suggesting a cellular mechanism.

Thanks for summarizing our findings and the suggestion. Unfortunately, eLIFE does not publish a graphical abstract.

(4) If Macrophages are responsible for IL1b release that via IL1R1 induces NETosis, what happens if you deplete macrophages? what is the role of epithelial cells?

Previous studies have found that macrophage depletion is protective in several models of ALI (Eyal. Intensive Care Med. 2007;33:1212–1218., Lindauer. J Immunol. 2009;183:1419–1426.), and other researchers have found that airway epithelial cells did not contribute to IL-1β secretion (Tang. PLoS ONE. 2012;7:e37689.). We have previously reported that epithelial cells produce IL-18 without LPS priming signal during LPS+HVV (Nosaka et al. Front Immunol 2020;11; 207). Thus, IL-18 is not sufficient to induce Hypoxemia as Saline+HVV treated mice do not develop hypoxemia (Nosaka et al. Front Immunol 2020;11; 207). We will now add this point to the revised discussion of the manuscript.

**Reviewer #2 (Public review):**
Summary:The manuscript by Nosaka et al is a comprehensive study exploring the involvement of IL1beta signaling in a 2-hit model of lung injury + ventilation, with a focus on modulation by hypothermia.Strengths:The authors demonstrate quite convincingly that interleukin 1 beta plays a role in the development of ventilator-induced lung injury in this model, and that this role includes the regulation of neutrophil extracellular trap formation. The authors use a variety of in vivo animal-based and in vitro cell culture work, and interventions including global gene knockout, cell-targeted knockout and pharmacological inhibition, which greatly strengthen the ability to make clear biological interpretations.

We thank the Reviewer for their positive feedback

Weaknesses:A primary point for open discussion is the translatability of the findings to patients. The main model used, one of intratracheal LPS plus mechanical ventilation is well accepted for research exploring the pathogenesis and potential treatments for acute respiratory distress syndrome (ARDS). However, the interpretation may still be open to question - in the model here, animals were exposed to LPS to induce inflammation for only 2 hours, and seemingly displayed no signs of sickness, before the start of ventilation. This would not be typical for the majority of ARDS patients, and whether hypothermia could be effective once substantial injury is already present remains an open question. The interaction between LPS/infection and temperature is also complicated - in humans, LPS (or infection) induces a febrile, hyperthermic response, whereas in mice LPS induces hypothermia (eg. Ganeshan K, Chawla A. Nat Rev Endocrinol. 2017;13:458-465). Given this difference in physiological response, it is therefore unclear whether hypothermia in mice and hypothermia in humans are easily comparable. Finally, the use of only young, male animals such as in the current study has been typical but may be criticised as limiting translatability to people.Therefore while the conclusions of the paper are well supported by the data, and the biological pathways have been impressively explored, questions still remain regarding the ultimate interpretations.

We agree with the reviewer that at two hours post LPS, there is only minimal pulmonary inflammation at that time (Dagvadorj et al Immunity 42, 640–653). This is a limitation to the experimental model we used in our study. Additionally, as the reviewer pointed out that LPS induces hyperthermia in human, but it is also well-established that physiological hypothermia occurs in humans with severe infections and sepsis (Baisse. Am J Emerg Med. 2023 Sep: 71: 134-138., Werner. Am J Emerg Med. 2025 Feb;88:64-78.). Therefore, the difference between human and mouse responses to sepsis or infections may be more nuanced. Furthermore, it is important to distinguish between physiological hypothermia (just <36°C) and therapeutic hypothermia (typically 32-34°C). We will add to the discussion whether hypothermia serves as a protective response, and the transition from normothermia to hyperthermia could have detrimental effects. We only used young male mice in our study as the Reviewer points out; we will also add this point to the revised discussion as a limitation of our study.

**Recommendations for the authors:**
(i) With hypothermia, metabolic activity would be expected to be reduced and therefore presumably impact on CO2/pH. These may have an impact on outcomes from ventilation, so could the authors include this data and discuss as appropriate?

We have now included these data in Suppl Fig 6. While we observed significant differences in blood pH and PaCO_2_ in Hypothermia treatment group, these values remained within clinically normal range (PaCO_2_ : 35 - 45 mmHg, pH : 7.35 - 7.45). Neither Alkalosis (PaCO_2_ < 35 mmHg , pH> 7.45) nor Acidosis (PaCO_2_ > 45 mmHg, pH < 7.35) was observed.

(ii) It is noticeable that there are quite large differences in experimental numbers between groups - typically 7-12, 5-12 in Figure 2. How were these N determined? For example is there a reason why there is apparently N = 8 for BALF neutrophils in the saline + HVV group (Figure 1c) but N = 12 for LPS + HVV group? Did any animals die during any of the protocols for example?

We conducted experiments with 4 mice per experiment (2 mice per group x2 or 4 mice per group) for ventilation experiments, and pooled data from 5-6 independent experiments or 3-4 independent experiments, respectively. No mouse mortality was observed (unless otherwise noted). However, in the severe ARDS group, some mice were dehydrated by the endpoint of experiments, preventing blood or BALF collections. As a result sample sizes were unequal in some case. Nevertheless, no data were selectively excluded.

(iii) Discussion - On page 13 you refer to data involving Cl-amidine administration. This does not seem to be related to any experiments reported in the manuscript.

We apology for this mistake and have removed it.

(iv) Methods - authors state that BALF was obtained after 150 minutes of ventilation, yet the experiments apparently lasted for 180 minutes. Presumably this is an error?

We apology for this inconsistency. We collected blood for measuring blood gas at 30 min and 150 min after ventilation. However, mice were kept on ventilator 30 min longer, and then mice were euthanized and BALF were collected. Thus, BALF were collected at 180 min, 30 minutes after the final blood draw. We have corrected the methods in revised manuscript.

(v) Statistical methods - authors state that sometimes Mann-Whitney U-test was used and sometimes unpaired t-test, presumably reflecting that some data were normally distributed and some were not. Could the authors please describe the tests used to confirm distribution of data.

We have clarified which stattistcal methods were used in our revised manuscript.

Briefly, Normality within the groups was assessed using the Shapiro-Wilk and KolmogorovSmirnov tests. Three-way ANOVA (Figure 1B; Supplemental Figure 1B-D; Supplemental Figure 6), one-way ANOVA (Supplemental Figure 4D-E; Supplemental Figure 5C), and two-way ANOVA were performed for data with more than two groups, followed by Tukey's post hoc test. Some groups analyzed by two-way ANOVA in Figure 1 and Supplemental Figure 1 failed the normality tests due to zero values (analyte not detected by ELISA) or the relatively small sample size, as samples were distributed across multiple measurements. However, the primary group of interest, LPS+HVV, showed significant differences from other groups with consistently low P-values in most datasets, supporting the decision to retain the ANOVA analyses. For comparisons between two groups, the Mann-Whitney U test was used when one or both groups failed the Shapiro-Wilk normality test, while the unpaired Student's t-test was applied to the remaining normally distributed data.